# Faster RCNN Target Detection Algorithm Integrating CBAM and FPN

**Wenshun Sheng \*, Xiongfeng Yu, Jiayan Lin and Xin Chen**

Pujiang Institute, Nanjing Tech University, Nanjing 211200, China
\* Correspondence: sws@njpji.edu.cn

**Abstract:** In the process of image shooting, due to the influence of angle, distance, complex scenes, illumination intensity, and other factors, small targets and occluded targets will inevitably appear in the image. These targets have few effective pixels, few features, and no obvious features, which makes it difficult to extract their effective features and easily leads to false detection, missed detection, and repeated detection, thus affecting the performance of target detection models. To solve this problem, an improved faster region convolutional neural network (RCNN) algorithm integrating the convolutional block attention module (CBAM) and feature pyramid network (FPN) (CF-RCNN) is proposed to improve the detection and recognition accuracy of small-sized, occluded, or truncated objects in complex scenes. Firstly, it incorporates the CBAM attention mechanism in the feature extraction network in combination with the information filtered by spatial and channel attention modules, focusing on local efficient information of the feature image, which improves the detection ability in the face of obscured or truncated objects. Secondly, it introduces the FPN feature pyramid structure, and links high-level and bottom-level feature data to obtain high-resolution and strong semantic data to enhance the detection effect for small-sized objects. Finally, it optimizes non-maximum suppression (NMS) to compensate for the shortcomings of conventional NMS that mistakenly eliminates overlapping detection frames. The experimental results show that the mean average precision (MAP) of target detection of the improved algorithm on PASCAL VOC2012 public datasets is improved to 76.2%, which is 13.9 percentage points higher than those of the commonly used Faster RCNN and other algorithms. It is better than the commonly used small-sample target detection algorithm.

**Keywords:** CF-RCNN; CBAM; FPN; target detection; attention mechanism

## 1. Introduction

The traditional machine learning method aims to train the classifier by manually dividing the area and filtering the data based on the existing specific data model. This method has a high time cost, and the obtained classifier does not perform well on datasets that are quite different from the specific data model, exposing the shortcomings of low robustness and weak generalization ability. An effective classifier not only needs to be able to continuously detect changes in the shape and state of the target, but also needs to be able to accurately predict and deal with system robustness problems caused by special changes.

The fusion of convolutional neural networks [1] in traditional machine learning models can better solve these problems. Because the convolutional neural network can independently learn and summarize the data characteristics of the target object, the selection and feature extraction of feature regions can be performed independently on the datasets of different scenarios, and it is applicable even in complex environments. After fusing the convolutional neural network, the target detection algorithm does not need manual intervention, which not only saves labor cost and strengthens the generalization ability, but also improves the system's robustness. Compared with the traditional machine learning manual

annotation method, it shows better performance in practical applications. Therefore, applying the convolutional neural network to assist target detection can enhance the ability of the model to extract image features and improve the accuracy of model object classification.

For the VGG16 network model [2] in object recognition applications, due to the shallow layer, the summary of object features is insufficient and the expression ability is weak. In order to solve such problems, a CF-RCNN algorithm integrating the CBAM (Convolutional Block Attention Module) attention mechanism [3] and FPN (Feature Pyramid Network) feature pyramid structure [4] is proposed. The algorithm uses the VS-ResNet network with a stronger expression ability and deeper layers to replace the VGG16 network in the feature extraction module of the traditional Faster RCNN algorithm, which strengthens the ability of the target detection model to analyze image features; VS-ResNet changes the ResNet-50 network [5] to the group convolution [6] mode, reducing the number of super parameters and the complexity of the algorithm model. The original residual structure is changed to an inverse residual structure [7], so that the $ReLU$ activation function [8] can better save function information. The risk priority number (RPN) is optimized to reduce the error rate of candidate frame filtering in the case of dense objects. Based on the traditional Faster RCNN algorithm, CF-RCNN fuses the FPN feature pyramid structure to improve the detection capability for small targets; the CBAM attention mechanism is introduced to focus on efficient information, to improve the accuracy of truncated or occluded object detection.

## 2. Related Works

In recent years, many scholars have performed active exploration in the field of small-scale target detection and truncated target detection and have achieved good results.

Salau et al. [9] presented a modified GrabCut algorithm for localizing vehicle plate numbers. The approach extends the use of the traditional GrabCut algorithm with the addition of a feature extraction method that uses geometric information to give accurate foreground extraction.

Yang et al. [10] proposed a multi-scale feature attention fusion network named parallel feature fusion with CBAM (PFF-CB) for occlusion pedestrian detection. Feature information of different scales can be integrated effectively into the PFF-CB module. The PFF-CB module uses a convolutional block attention module (CBAM) to enhance the important feature information in space and channels.

The authors in [11] proposed the DF-SSD algorithm, which is based on a dense convolutional network (DenseNet) for detection and uses feature fusion. It uses the algorithm framework of the Single Shot MultiBox Detector (SSD) [12] for reference and introduces DenseNet-S-32-1 to replace the original VGG16 network. In addition, DF-SSD also uses a multi-scale feature fusion mechanism. Their proposed residual prediction module is designed to enhance feature propagation along the feature extraction network, i.e., to use a small convolution filter to predict the target class and the offset of the bounding box position.

The authors in [13] proposed an improved double-head RCNN small target detection algorithm. A transformer and a deformable convolution module are introduced into ResNet-50, and a feature pyramid network structure CARAFE-FPN based on content perception feature recombination is proposed. In that regional recommendation network, the anchor generation scale is reset according to the distribution characteristics of the small target scale, so that the detection performance of the small target is further improved.

In [14], the authors proposed a BiFPN structure in the EfficientDet network, which is composed of a weighted bidirectional feature pyramid network, and added cross-scale connection to enhance the representation ability of features, and added corresponding weight to each input to better perform small target detection task.

In [15], the authors performed deep optimization based on the YOLOv5 architecture and proposed a multi-scale multi-attention target detection algorithm YOLO-StrVB is based on the STR (Swin Transformer) network structure. The method mainly aims at small target detection under a complex background, reconstructs the framework to build a multi-scale

network, increases the target detection layer, and improves the target feature extraction capability under different scales. Then, a bidirectional feature pyramid network is added for multi-scale feature fusion, and a jump connection is introduced. On this basis, the baseline backbone network end integrates STR architecture.

Fu et al. [16] proposed a deconvolutional single shot detector (DSSD) algorithm, in which ResNet with stronger learning ability is used as the backbone network, and a deconvolution layer is introduced to further reduce the missing rate of small targets.

Singh et al. [17] started from the perspective of training, considered the level of data, and selectively back-propagated the gradient of target instances with different sizes according to different changes in image scale, to realize scale normalization on an image pyramid (SNIP). While capturing the change in the object as much as possible, the model was efficiently trained with objects with appropriate proportions, and the detection performance was significantly improved.

Although the above algorithms improve the detection performance of the model to a certain extent, they do not fully consider the effect of local features on truncated target detection, and fail to optimize the light weight of the feature extraction module. The CF-RCNN algorithm proposed in this paper is based on group convolution, which significantly reduces the weight of the feature extraction network. The CBAM attention mechanism is introduced to fully consider local features. The FPN structure is integrated to improve the detection ability of small-sized targets and truncated targets. The CF-RCNN algorithm is superior to many of the most advanced methods. The specific advantages and disadvantages are compared in Table 1.

**Table 1.** Comparison of advantages and disadvantages of CF-RCNN and advanced algorithms.

| Detection Method | Advantage | Disadvantage |
|---|---|---|
| GrabCut algorithm | The proposed algorithm is not country-specific and can be used to detect LPs in complex environments | This method is not extended to motorcycle license localization |
| DF-SSD algorithm | The algorithm has a fast processing speed and a compact model | This algorithm is not applicable to large objects and objects without specific relationships |
| Scalable and Efficient Object Detection | This method uses fewer parameters and has a wide range of resource limits | The compound scaling method mentioned is complex and difficult to operate |
| Multi-Head Attention Detection | This method is suitable for small target detection in complex environments | There are obvious false detection and missing detection phenomena |
| SNIP | This method is effective in target recognition and detection under extreme scale changes | This method does not have universal generalization and the mAP value is low |
| CF-RCNN | This method has a strong detection ability for small-sized blocked or truncated targets, and has strong stability in complex environments | It is easy to cause the problem of multi-frame prediction of a single target |

The following are the primary contributions of this work:

- The ResNet-50 network model is optimized, and a VS-ResNet network model with stronger expression ability is proposed, which improves the classification accuracy in the target recognition process.
- The improved model of non-maximum suppression is applied to solve the problem of eliminating prediction frame errors in the process of classification and regression.
- A CF-RCNN algorithm is proposed and implemented for small-sized targets and truncated target detection.

## 3. Model Design and Implementation

### 3.1. Faster RCNN Model Design

Faster RCNN is a classic network proposed by Girshick in 2015 [18]. It consists of four modules, namely the feature extraction network module (Conv layers) [19], Region Proposal Network (RPN) [20,21] module, ROI Pooling [22] module, and Classification and Regression [23] module. The frame diagram of Faster RCNN is shown in Figure 1.

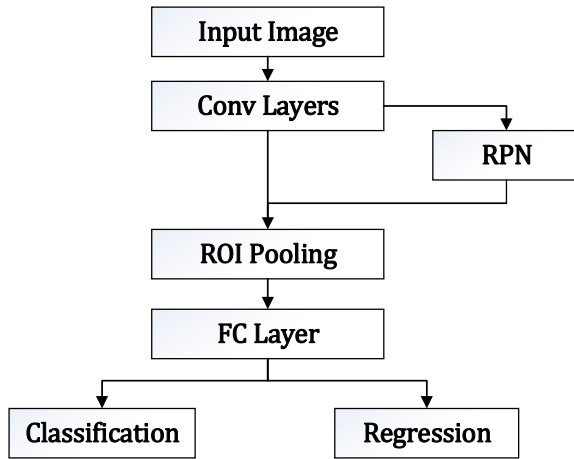

**Figure 1.** Schematic diagram of Fast RCNN algorithm framework.

After the original image is input, the shared convolution layer computes it. The results can be shared with the RPN. RPN makes region suggestions (about 300) on the feature map after convolutional neural network (CNN) convolution, extracts feature maps according to the region suggestions generated by RPN, performs ROI pooling on the extracted features, then classifies and regresses the processed data through the full connection layer.

The feature extraction network module [19] is an important part of the Fast RCNN algorithm. The flow of the feature extraction module is shown in Figure 2.

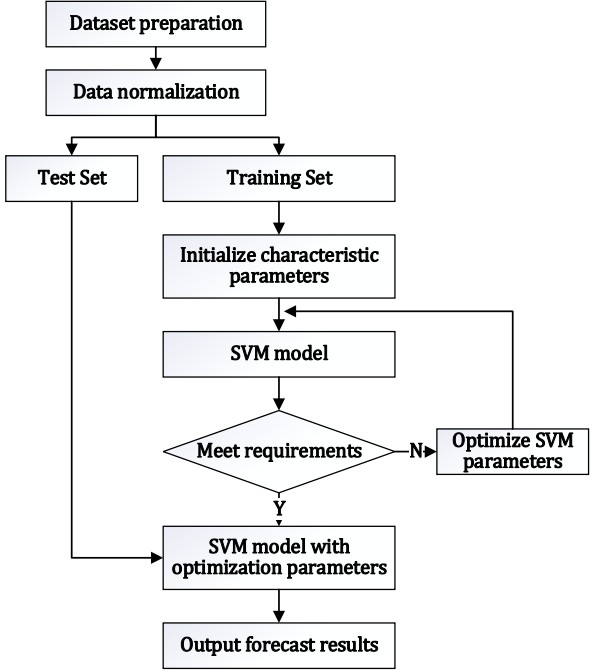

**Figure 2.** Feature extraction flow chart.

After normalization, the prepared dataset is divided into the training set and the test set. When the training set is trained by the Support Vector Machine (SVM) [24] model, better parameters are obtained by iteration, and finally, the SVM model meeting the requirements is obtained. The test dataset is input into the optimized SVM model, and the output result is finally obtained.

VGG16 is used as the feature extraction network in Faster RCNN. To break through the bottleneck of region selection in the predecessor target detection model, the new Fast RCNN algorithm model innovatively proposes the RPN algorithm, replacing the selective search (SS) [25] algorithm model used in Fast RCNN and RCNN, that is, the Faster RCNN algorithm can be understood as RPN + Fast RCNN. The SS algorithm is a simple image processing without a training function. RPN is a fully convolutional network, and its first several convolutional layers are the same as the first five layers of Faster R-CNN, so RPN can share the calculation results of convolutional layers to reduce the time consumption of region suggestions. The RPN module is used to replace the candidate region selection mode of the original SS algorithm, thus greatly reducing the time cost. RPN selects candidate regions according to image color, shape, and size, and usually selects 2000 candidate boxes that may contain identification objects.

When the detected image is input to the feature extraction network module, the feature image is generated after the module convolution operation, and it is input to the RPN module to select candidate regions. This module uses the center point of the sliding window as the anchor. All the pixels in the feature picture correspond to k anchor points, generating 128-, 256-, 512-pixel areas and 1:1, 1:2, and 2:1 scale windows, respectively. The combined result is a total of 9 windows, as shown in Figure 3.

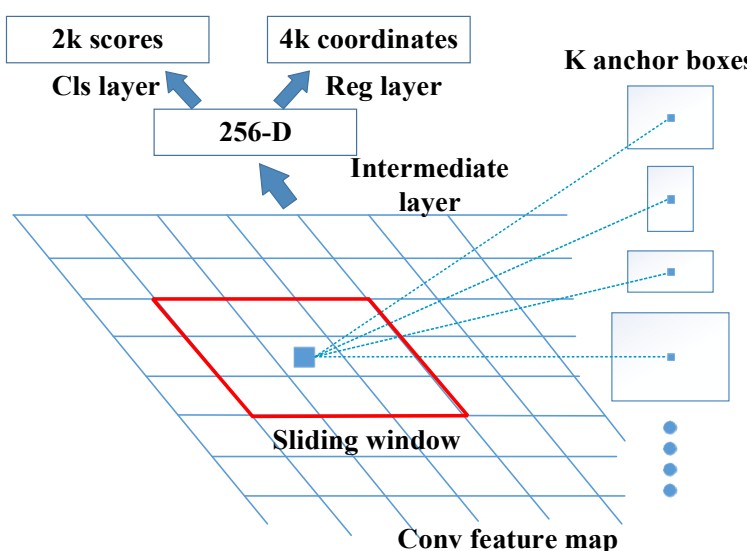

**Figure 3.** RPN structure diagram.

The generated candidate area and convolution layer feature map are input to the ROI Pooling module. ROI Pooling pools feature maps of different sizes into a uniform size, which facilitates their input to the next layer. Following the ROI Pooling module, the target classification and position regression module processes the data output from the ROI Pooling layer to obtain the object category of the candidate area and the modified image block diagram, and its processing formula can be expressed by Formula (1).

$$\begin{aligned}
\hat{G}_x &= P_w d_x(P) + P_x \\
\hat{G}_y &= P_h d_y(P) + P_y \\
\hat{G}_w &= P_w \exp(d_w(P)) \\
\hat{G}_h &= P_h \exp(d_h(P))
\end{aligned} \tag{1}$$

where $P_x$, $P_y$, $P_w$, and $P_h$ are the horizontal and vertical coordinates and the width and height values of the candidate box center pixel, respectively; $d_x$, $d_y$, $d_w$, and $d_h$ are the regression parameters of $N + 1$ categories of candidate boxes, with a total of $4 \times (N + 1)$ nodes; exp is an exponential function with the natural number $e$ as the base.

To solve the problem of weak expression ability of the VGG16 network due to its small number of layers, the VS-ResNet network is used to replace the VGG16 network. The VS-ResNet network is based on the ResNet-50 network improvement. In the original Faster RCNN, the CBAM attention mechanism and FPN feature pyramid structure are integrated to enhance the detection capability for small truncated or occluded objects. The specific improvement process is as follows.

### 3.2. Design of Group Convolution and Inverse Residual Structure

The structure of the packet convolution network refers to the multi-dimensional convolution [26] combination principle of Inception, which changes the single path convolution kernel convolution operation of the characteristic graph into a multi-channel convolution kernel convolution stack, reducing the parameters in the network and effectively reducing the complexity of the algorithm model, but its accuracy will not be greatly affected. VS-ResNet refers to the ResNeXt network [27] and uses 32 groups of 8-dimensional convolutional cores.

When a deep convolution network is used for convolution, the convolution parameter of the partial convolution kernel is 0, which results in partial parameter redundancy. The experiment proves that when the dimension is too low, too much image feature information of the ReLU activation function is lost. The ReLU function is shown in Formula (2).

$$\text{ReLU}(x) = \max(x, 0) = \begin{cases} x, x \geq 0 \\ 0, x < 0 \end{cases} \tag{2}$$

To solve this problem, the VS-ResNet network uses a reciprocal residual structure, with convolution kernel sizes of $1 \times 1$, $3 \times 3$, and $1 \times 1$, as shown in Figure 4.

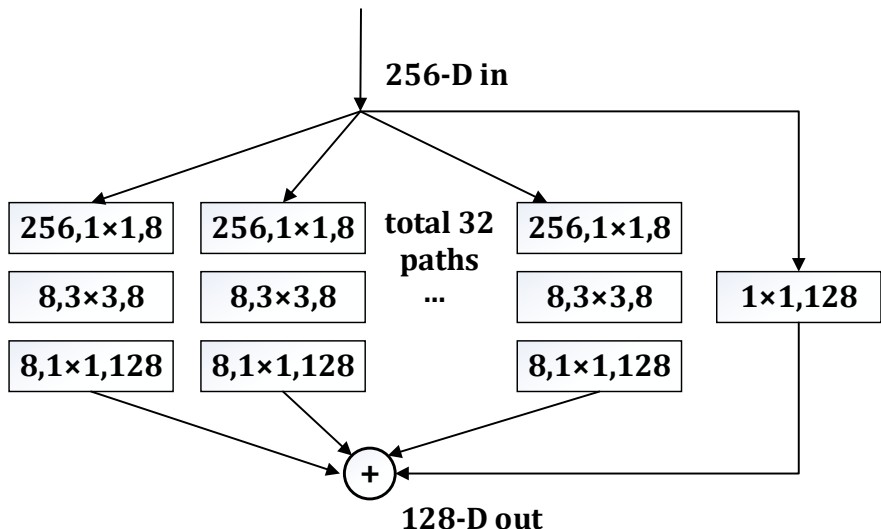

**Figure 4.** Convolution block structure diagram.

The common residual structure is first based on the output basis of the previous layer, using $1 \times 1$ convolution-kernel-size dimension reduction. Convolution kernels of $3 \times 3$ size are passed again to extract image features; lastly, dimension elevation of convolution size $1 \times 1$ is used. Unlike the ordinary residual structure, the inverted residual structure has exactly the opposite order of dimension increase and dimension reduction, that is, $1 \times 1$-size convolution is first used to increase the dimension. Additionally, you have a $3 \times 3$ convolution. Finally, a $1 \times 1$ convolution kernel is used to reduce the dimension to

the original feature map size. When the Re$LU$ function is activated linearly in the $x > 0$ region, it may cause the function value to be too large after activation, thus affecting the stability of the model. The Re$LU$ function does not limit output, allowing very high values on the positive side. However, Re$LU$6 limits the value of the positive side to 6, which can eliminate most of the value, so VS-RCNN uses the RE$LU$6 function instead of the Re$LU$ function, as shown in Formula (3).

$$ReLU6(x) = \min[\max(x, 0), 6] \tag{3}$$

### 3.3. Reference to Auxiliary Classifier

With the deepening of the network, the convergence of the feature extraction network becomes more difficult. It is necessary to retain a certain degree of the front-end network propagation gradient to alleviate the gradient disappearance. VS-ResNet adds an auxiliary classifier [28] after convolution layer 2 and convolution layer 4 of the ResNet-50 network, respectively to retain the low-dimensional output information of convolution layer 2 and convolution layer 4. In the final classification task, combined with the actual application, a fixed utility value is set and shallow feature reuse is used for auxiliary classification. The utility value of the auxiliary classifier in the VS-ResNet network is set to 0.1. The auxiliary classifier consists of an average pooling layer, a convolution layer, and two full connection layers, as shown in Figure 5. After the image is extracted by convolution layer 2 in ResNet-50, a feature map is generated and input to the first auxiliary classifier. Firstly, the number of network parameters is reduced by an average pooling downsampling layer, which uses a pool size of $14 \times 14$ with a step distance of 6. Then, the convolution layer is entered, which uses 128 convolution kernels of $1 \times 1$ with a sliding step of 1. The results are then flattened and fed into the fully connected layer that follows this layer.

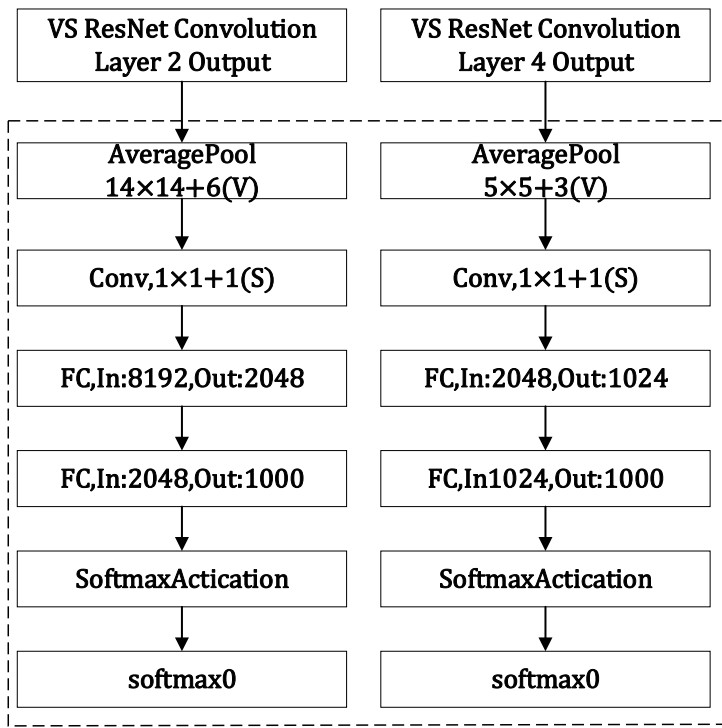

**Figure 5.** Auxiliary classifier diagram.

Dropout [29] at 50% is performed in two fully connected layers to prevent overfitting. Some parameters of the second auxiliary classifier are different from those of the first auxiliary classifier. The size of the pool is $5 \times 5$ instead of $14 \times 14$, and the step distance is changed from 6 to 3, which is based on the position of the auxiliary classifier in the convolutional neural network. The input of the two fully connected layers is 2048 and

1024 neurons, respectively. The output results of the two auxiliary classifiers are multiplied by the utility ratio set in VS-ResNet and then added to the final classification result. The addition of the auxiliary classifiers increases the gradient of network backpropagation and alleviates the phenomenon of gradient disappearance.

### 3.4. Detailed Structure of VS-ResNet Network

For the feature extraction network module, VS-ResNet changes the residual block structure from a funnel model to a bottleneck model, so that the activation function information can be better preserved, as shown in Table 2. Referring to the structural parameters of ResNet-50, the number of convolution channels of the first residual structure is changed from the original [64,64,256] to [256,256], and the last three residual structures are also modified successively from [128,128,512] to [512,512,256]. [256,256,1024] is [1024,1024,512] and [512,512,2048] is [2048,2048,1024]. The group convolution method is referenced in the inverted residual structure, and the number of groups is set to 32. By referring to the model of the Swin Transformer algorithm [30], the original hierarchical structure [3,4,6,3] is modified to [3,3,9,3], and auxiliary classifiers are added after the second and fourth layers to accelerate the convergence speed of the network to a certain extent and alleviate the phenomenon of gradient disappearance.

**Table 2.** Improved ResNet-50 network architecture.

| Network Layer | Output | 52 Layer |
|---|---|---|
| Convolution layer 1 | $112 \times 112$ | $7 \times 7, 64$, Step-length 2 |
| | | $3 \times 3$, Max pooling, Step-length 2 |
| Convolution layer 2 | $56 \times 56$ | $\begin{bmatrix} 1 \times 1, 256 \\ 3 \times 3, 256, C = 32 \\ 1 \times 1, 128 \end{bmatrix} \times 3$ |
| | | The first auxiliary classifier |
| Convolution layer 3 | $28 \times 28$ | $\begin{bmatrix} 1 \times 1, 512 \\ 3 \times 3, 512, C = 32 \\ 1 \times 1, 256 \end{bmatrix} \times 3$ |
| Convolution layer 4 | $14 \times 14$ | $\begin{bmatrix} 1 \times 1, 1024 \\ 3 \times 3, 1024, C = 32 \\ 1 \times 1, 512 \end{bmatrix} \times 9$ |
| | | The second auxiliary classifier |
| Convolution layer 5 | $7 \times 7$ | $\begin{bmatrix} 1 \times 1, 2048 \\ 3 \times 3, 2048, C = 32 \\ 1 \times 1, 1024 \end{bmatrix} \times 3$ |
| | $1 \times 1$ | Convolutional pooling layer, 1000 neurons, Softmax classifier |

### 3.5. Incorporate the CBAM Attention Mechanism

The attention mechanism can selectively ignore some inefficient information in the image, focus on the efficient information, reduce the resource consumption of the inefficient part, improve the network utilization, and enhance the ability of object detection. Therefore, the CBAM attention mechanism is integrated into the feature extraction network, and the channel attention mechanism and spatial attention mechanism are connected [31] to form a simple but effective attention module, whose structure is shown in Figure 6.

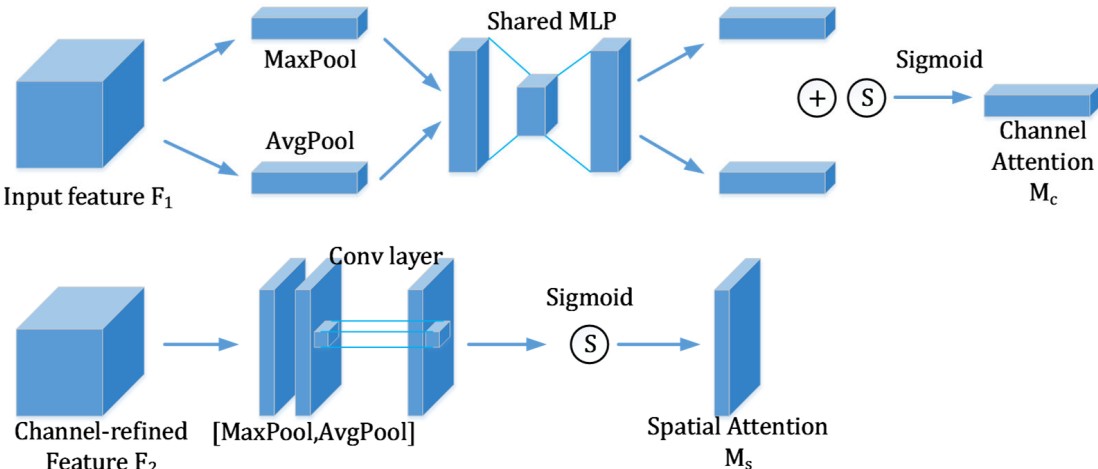

**Figure 6.** Schematic diagram of CBAM attention mechanism.

In the channel attention module, the global average pooling and maximum pooling of the same input feature space are performed to obtain the spatial information of the feature map, and then the obtained feature space information is input into the multi-layer awareness mechanism module of the next layer for dimension reduction and dimension-increase processing. The weight of the two shared convolution layers in the multi-layer awareness network is shared. Then, the characteristics of the perceptual network output are added and then processed with the sigmoid activation function to obtain channel attention. The calculation formula is shown in Formula (4).

$$Mc(F) = \varepsilon[MLP(F_{avg}^c) + MLP(F_{\max}^c)] \tag{4}$$

Among them, *Mc* is the channel attention module calculation factor, $\varepsilon$ is the sigmoid activation function, *MLP* is the multi-layer perceptron, and *F* represents the feature vector.

Spatial attention features are complementary to channel attention and reflect the importance of input values in spatial dimensions. The calculation formula is shown in Formula (5). First, global average pooling and global maximum pooling of one channel dimension are performed on the feature map, then the two features are spliced, and finally the dimension is reduced to one by $7 \times 7$ convolution post-channel processing using the sigmoid function [32] to generate spatial attention feature maps.

$$Ms(F) = \varepsilon\left\{conv_{7\times 7}[unit(F_{avg}^s, F_{\max}^s)]\right\} \tag{5}$$

Among them, *Ms* is the space attention module calculation factor, $\varepsilon$ is the sigmoid activation function, *MLP* is the multi-layer perceptron, *F* represents the feature vector, *unit* is the channel combination, and *conv* is the convolution operation.

In order to facilitate the use of pre-trained models during the experiment, CBAM is not embedded in all convolutional residual blocks, but only acts after different convolutional layers.

### 3.6. Introduce FPN Feature Pyramid Structure

In order to alleviate the unsatisfactory detection ability of the Faster RCNN algorithm for small-sized targets, the FPN feature pyramid network model is introduced into the Faster RCNN target detection algorithm. The FPN network model is divided into two network routes. One of the network routes produces multi-scale features from bottom to top, connecting the high-level features with high semantics and low resolution and the low-level features with high resolution and low semantics [33]. Another network route is from top to bottom; after some layer changes, the rich semantic information contained in the upper layer is transferred layer by layer to the low-layer features for fusion [34].

Compared with the SSD algorithm, FPN also uses multi-level features and multi-scale anchor frames. SSD predicts the low-level data alone, where it is difficult to ensure strong semantic features, and the detection effect is not ideal for small targets.

Figure 7 is a schematic diagram of the FPN feature pyramid structure. In the figure, the bottom-up feature map with computed order on the left is {C2, C3, C4, C5}, and the top-down feature pyramid structure on the right is {P2,P3,P4,P5}. The CF-RCNN algorithm uses the above VS-ResNet as the backbone extraction network of the FPN feature pyramid structure. The image part on the left side of Figure 7 is a downsampling model. During the feature extraction operation of this model, the value of the step size is set as {4,8,16,32}. In the upsampling model of the right image, the upper feature map is convolved with a convolution kernel of $1 \times 1$ size, the step size is set to 1, and the number of channels is 256, to adjust the dimension to be consistent so that it can be fused with the lower feature. Then, after $3 \times 3$ size convolution, the aliasing situation in the 2-fold upsampling process is eliminated to obtain the feature map. {P2, P3, P4, P5} share the weight of RPN and Fast RCNN, and use a different anchor size of {$32^2$, $64^2$, $128^2$, $256^2$} and the anchor ratio of {1:2, 1:1, 2:1} on the {P2, P3, P4, P5} feature map to select candidate boxes.

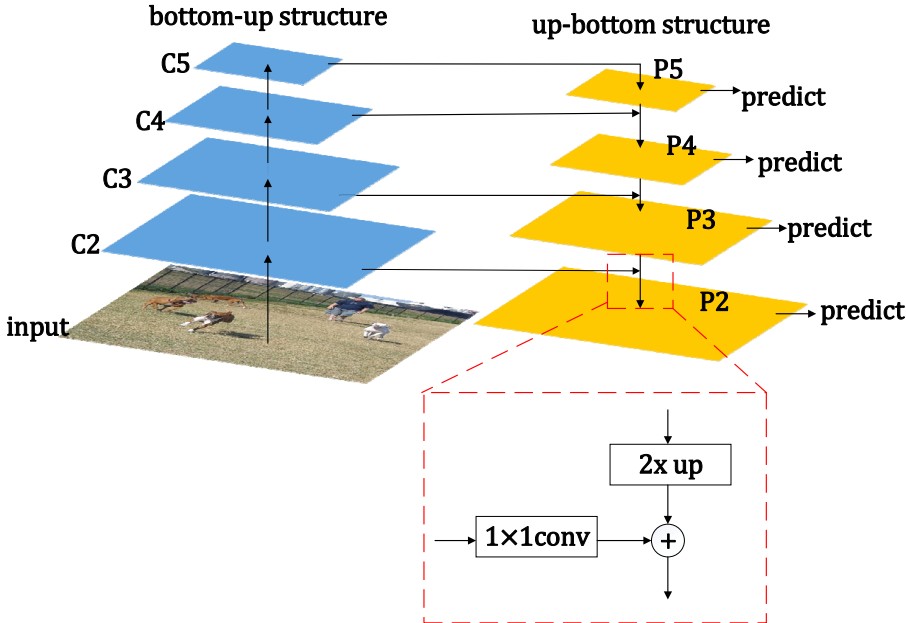

**Figure 7.** FPN Feature Pyramid Structure.

### 3.7. Improvement of Non-Maximum Suppression NMS Algorithm

Non-Maximum Suppression (NMS) is an important part of the target detection algorithm, and its main function is to eliminate redundant candidate boxes generated in the RPN network. The elimination criterion is shown in Formula (6).

$$S_i = \begin{cases} S_i, IOU(M, b_i) < N_t \\ 0, IOU(M, b_i) \geq N_t \end{cases} \tag{6}$$

where $S_i$ is the score of the $i$ th candidate box; $M$ indicates the candidate box with the largest score; $b_i$ is the candidate box to be scored; $IOU$ is the combined ratio of $b_i$; and $N_t$ is the threshold value of $IOU$ set in NMS.

When the ratio between the candidate box to be scored and the candidate box with the largest score exceeds the threshold, the NMS deletes the candidate box to be scored. As a result, NMS mistakenly deletes overlapping candidate boxes in partially crowded

and truncated complex scenarios. To solve this problem, CF-RCNN uses the Soft-NMS algorithm. The Soft-NMS score function is shown in Formula (7).

$$S_i = \begin{cases} S_i, IOU(M, b_i) < N_t \\ S_i(1 - IOU(M, b_i)), IOU(M, b_i) \geq N_t \end{cases} \tag{7}$$

If the combined ratio is less than the threshold, the score remains unchanged. If it is greater than or equal to the threshold, $S_i$ is the difference between $S_i$ multiplied by 1 and the combined ratio. Compared with NMS, Soft-NMS does not directly overlap candidate boxes, but resets $S_i$ to a smaller score.

### 3.8. Overall Structure of CF-RCNN

Combined with the above improvement measures, the detailed structure of the CF-RCNN algorithm is shown in Figure 8.

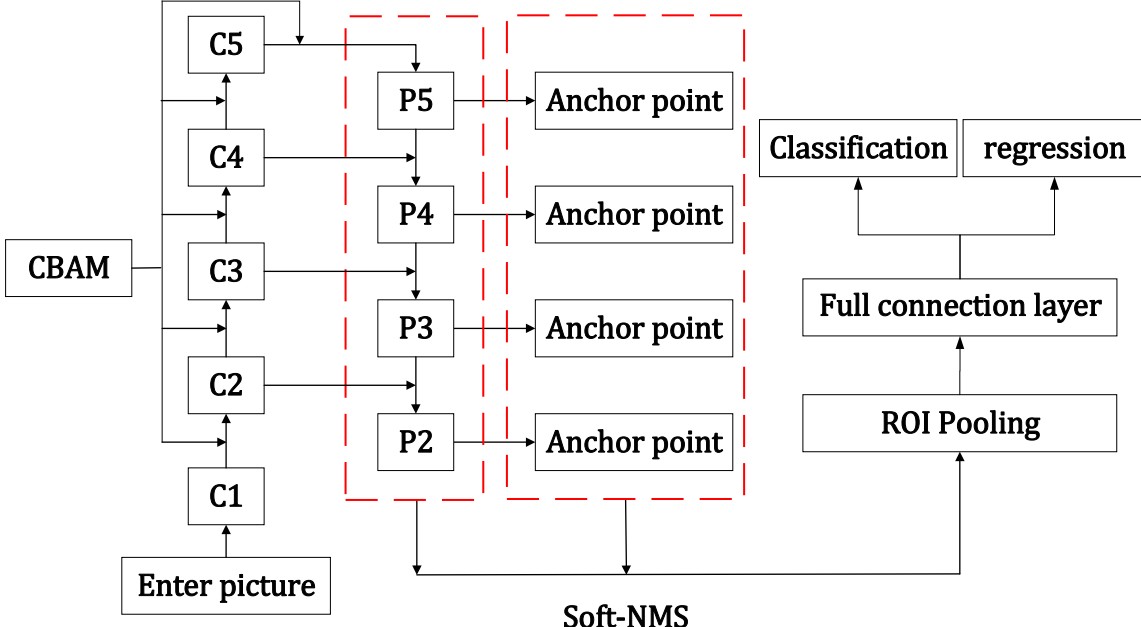

**Figure 8.** Overall structure of CF-RCNN.

Firstly, CF-RCNN takes the image stream as input, extracts image features based on the inverted residual ResNet-50 model, and obtains intermediate feature vector graphics. Secondly, the CBAM attention mechanism is used to calculate the attention weight in the two dimensions of space and channel, and adjusts the parameters of the intermediate feature map. Third, based on the FPN feature pyramid structure, the feature map is multi-scaled. Fourth, on each feature map, the RPN network uses a single scale to select regions that may contain objects and balances the positive and negative sample ratios through Soft-NMS. Finally, the classification and position regression operations are performed on the objects in the selected area.

## 4. Experimental Results and Analysis

The experiment is based on the deep learning network framework PyTorch 1.81, the programming language is Python 3.8, the CPU configuration is a 10-core Intel(R) Xeon(R) Gold 5218R CPU @ 2.10 GHz, and the memory is 64 GB. The GPU is configured as an RTX 3090 + CUDA 1.1, with 24 GB of video memory. The operating system uses Ubuntu 18.04 version.

### 4.1. Public Dataset Preparation

In order to verify the effectiveness of the improvement measures in CF-RCNN for improving the performance of object detection, object recognition and object detection experiments are performed on the CIFAR-10 dataset [35] and the PASCAL VOC2012 dataset [36], respectively. The CIFAR-10 dataset has a total of 60,000 images, which are divided into 10 types of objects to be recognized. Each type of object to be recognized has a capacity of 6000 images, and the ratio of the training set and test set is 5:1. The PASCAL VOC2012 dataset contains 20 detection categories and a total of 17,125 images, of which 5718 images are used in the training set and 5824 images are used in the validation set.

### 4.2. Object Recognition Experiment Results Analysis

The experiment compares the change in the error rate of the VGG16, ResNet-50, and VS-ResNet networks on the CIFAR-10 dataset with the increase in the number of iterations. The number of iterations is set to 35, the training batch is 16, and the learning rate is set to 0.0001. The experimental results are shown in Table 3 and Figure 9.

**Table 3.** Comparison of the lowest error rates of different convolutional networks.

| Network Model | Number of Plies | Minimum Error Rate |
|---|---|---|
| VGG16 | 16 | 0.1106 |
| ResNet-50 | 50 | 0.0941 |
| VS-ResNet-50 | 52 | 0.0809 |

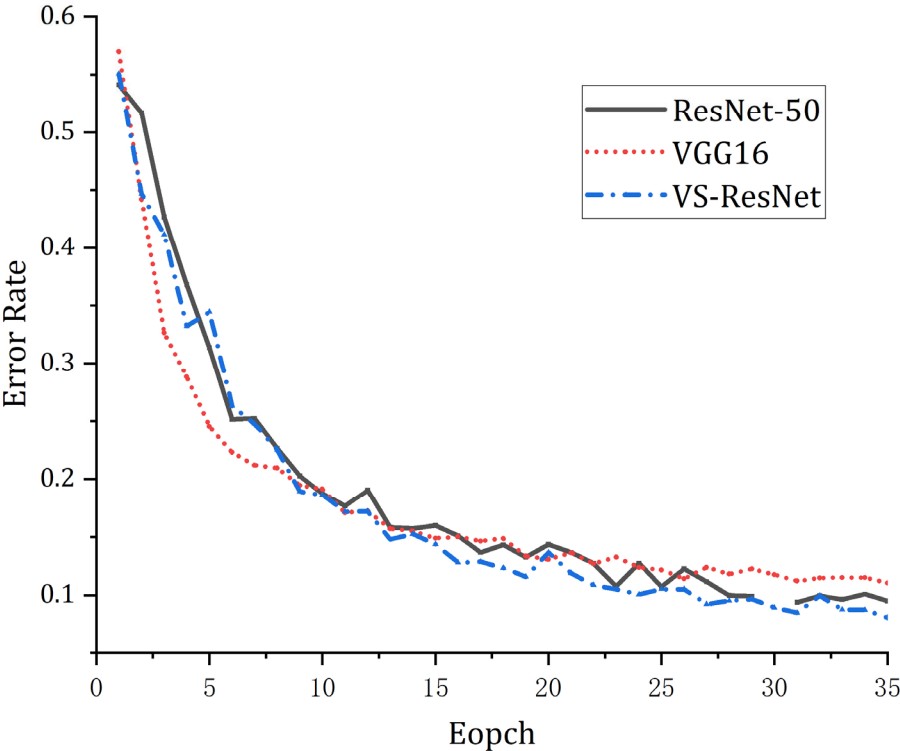

**Figure 9.** Comparison of error rates of different convolutional networks.

The data in Table 3 show that the ResNet-50 network shows better performance in the object classification task than the traditional VGG16 network, with an increase of 1.65 percentage points in the accuracy rate. Compared with the initial ResNet-50, the VS-ResNet network increases by 1.32 percentage points, and it increases by 2.97 percentage points compared with the VGG16 model. In Figure 9, the abscissa is the number of iterations, and the ordinate is the test error rate. It can be seen from Figure 9 that the VGG16 model has a certain advantage in the convergence speed, and the advantage is obvious in the first

10 iterations. After 15 iterations, the descending gradient of the model becomes smaller and gradually tends to saturation. The VS-ResNet model alleviates the problem of the slow convergence speed of ResNet-50 to a certain extent. The convergence improvement effect of the first 5 iterations is obvious, and the classification error rate after 15 iterations has obvious advantages over ResNet-50 and VGG16. To sum up, the VS-ResNet network accelerates the convergence speed of ResNet-50 and reduces the classification error rate.

### 4.3. Analysis of Target Detection Experiment Results

In order to verify the impact of different optimization strategies on the performance of the algorithm, the target detection experiment is based on the traditional Faster RCNN algorithm, combined with the above optimization measures.

#### 4.3.1. AP$_{50}$ Detection Accuracy Comparison

The experiment sets the initial learning rate to 0.005, the momentum parameter to 0.9, and the weight decay coefficient to 0.0005. AP$_{50}$ refers to the detection accuracy when the IOU threshold is 0.5 [37,38]. The batch size is 16 and the number of iterations is 20 k. The experimental results are shown in Table 4.

**Table 4.** Impact of different optimization strategies.

| VGG16 | VS-ResNet | CBAM | FPN | Soft-NMS | Detection Accuracy AP$_{50}$ (%) | Average Processing Time (ms) |
|:---:|:---:|:---:|:---:|:---:|:---:|:---:|
| √ | | | | | 62.3 | 275 |
| | √ | | | | 65.6 | 283 |
| | √ | √ | | | 69.9 | 292 |
| | √ | √ | √ | | 74.5 | 311 |
| | √ | √ | √ | √ | 76.2 | 316 |
| √ | | √ | √ | √ | 72.4 | 302 |

Note: √ indicates that the optimization strategy uses this module.

#### 4.3.2. AP Detection Accuracy Comparison

Comparing the data in Table 4, it can be seen that compared with Faster RCNN, CF-RCNN's detection accuracy AP$_{50}$ increases by 13.9 percentage points to 76.2%, and the average processing time is slightly lower than that of Faster RCNN, but it can still meet the real-time requirements. In order to verify the impact of the improvement measures on the detection ability of small-sized objects, objects smaller than $32 \times 32$ pixels are selected as the experimental dataset to compare the detection accuracy. The AP value is the average AP accuracy achieved for 10 different IOU thresholds of 0.50:0.05:0.95. The results are shown in Table 5.

**Table 5.** Effect of FPN structure on fast RCNN performance.

| Algorithm | AP (%) |
|:---:|:---:|
| Faster RCNN | 5.9 |
| Faster RCNN + FPN | 19.8 |
| Faster RCNN + FPN + Soft-NMS | 21.3 |

It can be seen from Table 5 that after FPN and Soft-NMS are introduced into the Faster RCNN model, the AP value for small-sized object detection is increased by 15.4 percentage points, which alleviates the weak detection performance of the target detection algorithm based on small-sized objects.

#### 4.3.3. Truncated Objects Detection Accuracy Comparison

To verify whether the improved algorithm optimizes the detection ability of the original algorithm for truncated targets, the picture is manually edited, some detected

objects are intercepted, and CF-RCNN and Faster RCNN are used to detect them. The test results are shown in Figure 10. The experimental results show that the accuracy of CF-RCNN is higher than that of Fast RCNN, and the accuracy of the image frame and the possibility of classification are higher.

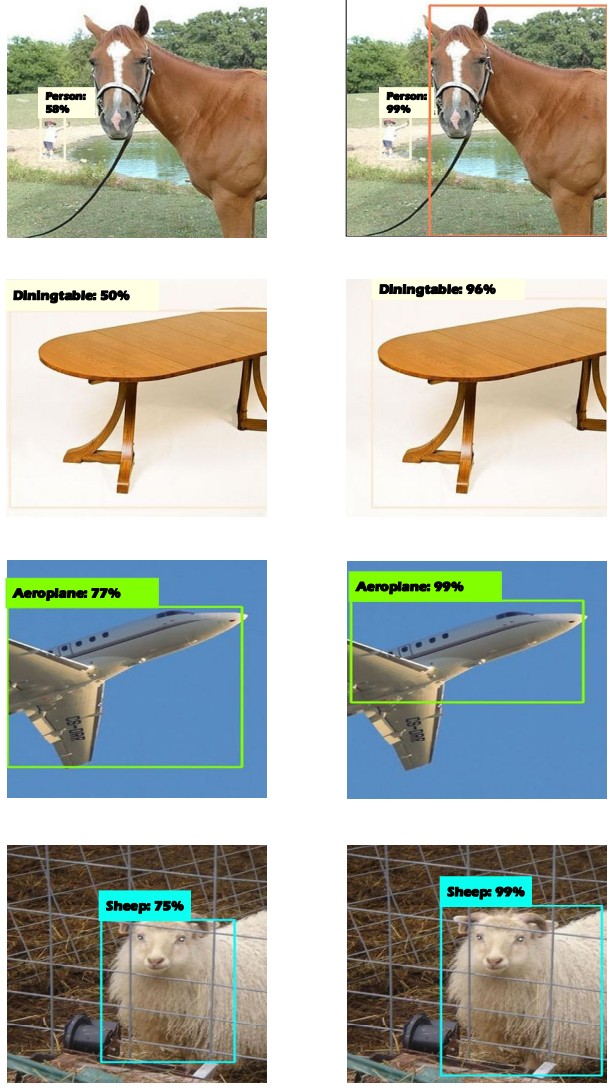

**Figure 10.** Comparison of truncated target detection results.

In the first row, CF-RCNN identifies a horse with only half a body, but Faster RCNN does not do this. The recognition rate of the pictures in other corresponding picture groups is also improved. The detection results of the third row of aircraft show that CF-RNN can not only effectively identify small truncated targets, but also identify large-size targets, and the recognition of the aircraft body is more complete.

### 4.3.4. Recall Rate Comparison

The data in Table 6 are the recall rate changes on the PASCAL VOC2012 dataset before and after algorithm optimization. Average Recall (AR) is the average recall rate, which refers to taking the maximum recall rate for different IOUs and then calculating the average [39]. The experimental data show that the recall rate of the improved algorithm on PASCAL VOC 2012 changes from the original 53.9% to 62.8%, which increases by 9.3 percentage points.

**Table 6.** Comparison of recall rate before and after algorithm optimization.

| Algorithm | AR (%) |
|---|---|
| Faster RCNN | 53.9 |
| CF-RCNN | 62.8 |

### 4.3.5. Algorithm Efficiency Comparison

Different classical algorithms and CF-RCNN algorithms are used for comparative experiments, and the $AP_{50}$ values and average processing time changes of several algorithms are compared. The experiments are based on the same dataset to ensure that the experimental environment of several algorithms is consistent. As shown in Table 7, the classical algorithms of the experiment include YOLOv5 [40] and SSD algorithms.

**Table 7.** Performance comparison between classical target detection algorithm and CF-RCNN algorithm.

| Algorithm | $AP_{50}$ (%) | Average Processing Time (ms) |
|---|---|---|
| YOLOv5 | 72.9 | 84 |
| SSD | 60.8 | 91 |
| CF-RCNN | 76.2 | 111 |

### 4.3.6. Adaptability Comparison

Compared with the other two classical target detection algorithms, CF-RCNN has a longer processing time, but meets the basic real-time requirements, and the detection accuracy is higher than the other two algorithms.

In order to test the adaptability of the CF-RCNN algorithm to the truncation and occlusion of objects, the images with object occlusion or truncation are screened out on the PASCAL VOC 2012 dataset, and Faster RCNN and CF-RCNN are used for comparative experiments. Figure 11 shows part of the experimental results.

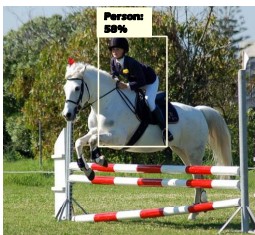
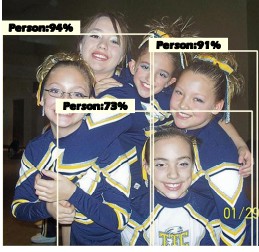
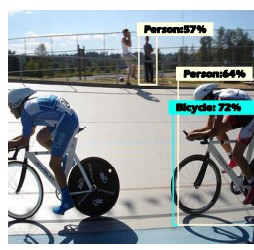
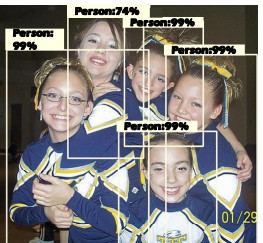
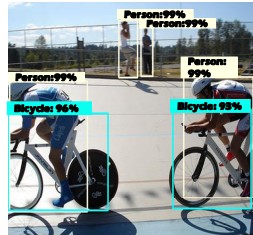

**Figure 11.** *Cont.*

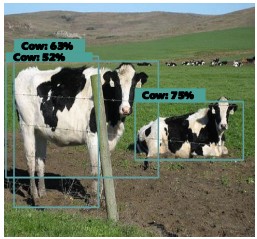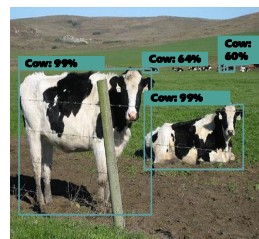
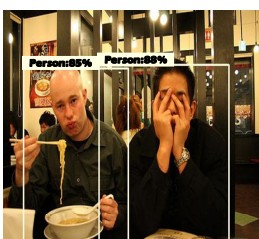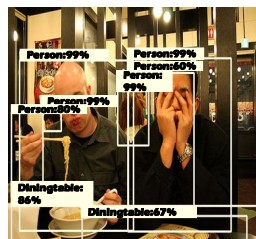

**Figure 11.** Comparison of detection results before and after algorithm optimization.

From the comparison of the experimental results in Figure 11, it can be seen that the traditional Faster RCNN algorithm can basically meet the detection requirements of truncated or occluded objects in a simple environment. The results of object classification are not significantly different from those of the CF-RCNN algorithm, but they are obviously inferior to the CF-RCNN algorithm in object location selection. In the case of truncation or occlusion of objects with complex scenes, traditional Faster RCNN has most false detection and missed detection phenomena; the CF-RCNN algorithm makes up for such defects well, and better alleviates the unsatisfactory detection results in the case of occlusion or truncation.

## 5. Conclusions

In recent years, more and more scholars have entered the field of computer vision and artificial intelligence for exploratory research, which has led to considerable development of the technology in this field. The original Faster RCNN algorithm is not ideal for the detection of small-sized objects and occluded or truncated objects. The improved CF-RCNN algorithm solves this problem well and makes up for the defects. Compared with the VGG16 network, the VS-ResNet network not only retains the characteristics of the VGG16 network with fast iterative convergence speed but also improves the recognition accuracy of the network. The introduction of the FPN and CBAM modules has greatly enhanced the detection ability of the algorithm for small-sized occluded or truncated objects. Experiments show that not only can the CF-RCNN algorithm achieve better detection results in simple scenes, but it also has a better stability based on complex environments. However, it is easy to cause the problem of multiple prediction frames for a single object. The implementation process of the algorithm is detailed, but it also increases the time complexity of the algorithm. The next step will explore and study the method of cascading multiple detectors to solve this problem.

**Author Contributions:** Conceptualization, W.S. and X.Y.; methodology, W.S. and X.Y.; investigation, X.Y. and J.L.; software, J.L. and X.C.; supervision, W.S. and X.Y.; writing—original draft preparation, J.L. and X.C.; writing—review and editing, W.S., X.Y. and X.C.; project administration, W.S., X.Y. and J.L. All authors have read and agreed to the published version of the manuscript.

**Funding:** This work was sponsored by the Key University Science Research Project of Jiangsu Province (19KJD520005), Qinglan Project of Jiangsu Province of China (Su Teacher's Letter (2021) No. 11), and the Young teacher development fund of Pujiang Institute Nanjing Tech University ((2021) No.73).

**Institutional Review Board Statement:** No research involving humans or animals.

**Informed Consent Statement:** Informed consent was obtained from all subjects involved in the study.

**Data Availability Statement:** Data are available on request due to restrictions, e.g., privacy or ethical.

**Acknowledgments:** The authors would like to thank the editor and the anonymous reviewer whose constructive comments will help to improve the presentation of this paper.

**Conflicts of Interest:** The authors declare no conflict of interest.

## Abbreviations

| | |
|---|---|
| RCNN | Region Convolutional Neural Network |
| CBAM | Convolutional Block Attention Module |
| FPN | Feature Pyramid Networks |
| CF-RCNN | Region convolutional neural network integrating convolutional block attention module and feature pyramid networks algorithm |
| NMS | Non-Maximum Suppression |
| MAP | Mean Average Precision |
| VGG16 | VGG16 network model |
| VS-ResNet | VS-ResNet network |
| ResNet-50 | ResNet-50 network |
| RPN | Risk Priority Number |
| GrabCut | GrabCut algorithm |
| PFF-CB | Parallel Feature Fusion with CBAM |
| DenseNet | Dense Convolutional network |
| ResNeXt | ResNeXt network |
| AP | Average Precision |
| AR | Average Recall |
| CNN | Convolutional Neural Network |

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
