# Peer review of "Faster RCNN Target Detection Algorithm Integrating CBAM and FPN"

_applsci, doi:10.3390/app13126913_

Round 1

Reviewer 1 Report

In this paper, the authors highlight the amalgamation of CBAM and FPN to propose the faster RCNN target detection algorithm to improve the detection ability in case of obscured or truncated objects. But, there are following major concerns associated with the paper that authors need to address, which is mentioned as follows:

a) The abstract is very vague. It is not clear why there is a need to propose a RCNN target detection algorithm. There is no motivation and goal highlighted for the proposed system  in the abstract section. The authors are requested to rewrite the entire abstract section with clear motivation and contribution of the proposed system. Moreover, experimental results mentioned should be explained more clearly.

b) Authors have not included the literature survey section in the paper, then how authors are going to show that their proposed system is better than the traditional methods? Also, how they are going to identify parameters of the proposed system that surpasses the traditional method in terms of performance. They should show it with the help of a  comparative analysis table.

c) In the entire paper, there are no abbreviations defined for the terms used in the paper. For example, RCNN, CBAM, FPN, etc., have no definition. Then, how readers can identify these abbreviations. Also, authors should prepare a table containing all the abbreviations along with their definitions.

d) In section 3.1, why authors replace ReLU (x) with ReLU6(x) in equation 2 and 3? They should explain it more concisely. Moreover, variables are not defined in equation 4 which makes it difficult to understand for the readers. Also, authors should highlight the novelty of their proposed approach, otherwise, their work can't be considered to be appropriate.

e) In section 4.3, the authors have performed target experiment detection results to verify the impact of different optimization strategies based on the traditional faster RCNN algorithm. in terms of accuracy. But, there is no graphical representation to show the accuracy comparison. Only tabular comparison is not sufficient.

f) In conclusion, the authors need to highlight the motivation and contribution of the proposed system. Also, experimental results are not defined properly, for example, what accuracy authors are obtaining for the proposed system. Moreover, the proposed system is only applicable for small-sized occluded or truncated objects. Then, how authors are going to implement it for large-size occluded or truncated objects. There is no future work defined for the proposed system, which makes it difficult for researchers to work on the proposed system in the future. 

Reviewer 2 Report

1. Please mention the number of training parameters in fastRCNN vs proposed algorithm, training time required.

2. Author may try to present the proposed algorithm flowchart simillar like figure1 that could help reader quick understand the changes in model.

Reviewer 3 Report

The authors proposed a model to detect the objects of images. They integrated CBAM (Convulation Block Attention Module) and FPN (Feature Pyramid Network) with RCNN to find the small object in an image. The article is well written. I have few suggestions

1. For first-time use, please use the full form of all abbreviations (in the Abstract too). 

2. Other compare different Net, it would be appreciable if the author can compare their best method with other state-of-art methods. 

Round 2

Reviewer 1 Report

The authors have incorporated all suggested comments. It is accepted from my side. 
